# Fine-Grained Analysis of Stability and Generalization for Modern Meta Learning Algorithms

Jiechao Guan[1,3]          Yong Liu[2,3]          Zhiwu Lu[2,3,*]

[1] School of Information, Renmin University of China, Beijing, China
[2] Gaoling School of Artificial Intelligence, Renmin University of China, Beijing, China
[3] Beijing Key Laboratory of Big Data Management and Analysis Methods
{2014200990, liuyonggsai, luzhiwu}@ruc.edu.cn

## Abstract

The support/query episodic training strategy has been widely applied in modern meta learning algorithms. Supposing the $n$ training episodes and the test episodes are sampled independently from the same environment, previous work has derived a generalization bound of $O(1/\sqrt{n})$ for smooth non-convex functions via algorithmic stability analysis. In this paper, we provide fine-grained analysis of stability and generalization for modern meta learning algorithms by considering more general situations. Firstly, we develop matching lower and upper stability bounds for meta learning algorithms with two types of loss functions: (1) nonsmooth convex functions with $\alpha$-Hölder continuous subgradients ($\alpha \in [0,1)$); (2) smooth (including convex and non-convex) functions. Our tight stability bounds show that, in the nonsmooth convex case, meta learning algorithms can be inherently less stable than in the smooth convex case. For the smooth non-convex functions, our stability bound is sharper than the existing one, especially in the setting where the number of iterations is larger than the number $n$ of training episodes. Secondly, we derive improved generalization bounds for meta learning algorithms that hold with high probability. Specifically, we first demonstrate that, under the independent episode environment assumption, the generalization bound of $O(1/\sqrt{n})$ via algorithmic stability analysis is near optimal. To attain faster convergence rate, we show how to yield a deformed generalization bound of $O(\ln n/n)$ with the curvature condition of loss functions. Finally, we obtain a generalization bound for meta learning with dependent episodes whose dependency relation is characterized by a graph. Experiments on regression problems are conducted to verify our theoretical results.

## 1 Introduction

The last decade has witnessed the success of deep learning techniques in machine learning community [28, 25, 11]. However, the need of large amount of annotated data hinders their application in real-life scenarios. To alleviate this issue, meta learning [4], which employs knowledge from past tasks to facilitate adaptation to the new task, has emerged as a promising direction to reduce annotation cost.

Traditional meta learning algorithms directly minimize the empirical error over all samples in the training tasks [3, 34, 35, 38, 36]. To improve the generalization ability of meta learning algorithms, recent works propose the support/query (S/Q) episodic training strategy [45, 20, 41]. Specifically, in modern meta learning algorithms, each episode/task is split into two non-overlapped parts: support set and query set. The support set is used to learn a hypothesis, and the query set is used to measure the performance of the learned hypothesis on that episode. Therefore, the S/Q episodic strategy regards

---

*Corresponding Author

each task as a training instance and updates the meta learning model by implementing episode-level stochastic gradient descent (SGD). Supposing the $n$ training episodes and the test episodes are sampled independently from the same environment, previous work [8] has derived a high-probability generalization bound of $O(1/\sqrt{n})$ for modern meta learning. Such bound is obtained via algorithmic stability analysis [6] for smooth non-convex loss functions. However, it is still unknown whether such generalization bound of $O(1/\sqrt{n})$ is optimal, and whether we can obtain sharper bounds for modern meta learning. Further, there is still lack of comprehensive comparisons between the bounds obtained via S/Q episodic training and the bounds obtained via traditional empirical risk minimizing (ERM).

In this work, we will address the above problems via algorithmic stability analysis. Algorithmic stability, roughly speaking, bounds the change in the model output by the algorithm when a single data in the dataset is replaced. Our goal is to provide fine-grained analysis of stability and generalization for modern meta learning algorithms by considering more general situations. Firstly, we develop matching lower and upper stability bounds for meta learning algorithms with two types of loss functions: **(1)** nonsmooth convex functions with $\alpha$-Hölder continuous subgradients where $0 \le \alpha < 1$; **(2)** smooth (including convex and non-convex) functions. Our tight stability bounds demonstrate that, in the nonsmooth convex case, modern meta learning algorithms can be less stable than in the smooth convex case. In particular, the lower stability bound for nonsmooth convex functions is vacuous even if we train modern meta learning algorithms with a relatively small constant step size in SGD. In the smooth non-convex case, our derived bound is sharper than the existing one [8], especially in the setting where the number of SGD iterations is larger than the number $n$ of training episodes. Secondly, we provide high-probability generalization bounds for modern meta learning algorithms with the aforementioned two types of loss functions. Specifically, we first demonstrate that, under the independent episode environment assumption, the bound of $O(1/\sqrt{n})$ is near optimal and is independent of the sample size $m$ per episode. We thus show that, in terms of the sharpness of the generalization bounds, the S/Q episodic training strategy is superior to the traditional ERM strategy for meta learning (see Remark 6). To obtain faster convergence rate, we next show how to yield a deformed generalization bound of $O(\ln n/n)$ with additional curvature assumption (i.e., Polyak-Łojasiewicz condition [46]) of the loss function. Finally, we use the graph approximation technique [47] to obtain a bound for meta learning with dependent episodes whose dependency relation can be characterized by a graph. To the best of our knowledge, this is the first bound that captures how the dependency between episodes can affect the generalization behavior of meta learning algorithms.

Overall, our contributions are four-fold: **(1)** We provide matching lower and upper stability bounds for modern meta learning algorithms with general loss functions. The stability bound for nonsmooth convex functions implies that modern meta learning algorithms are not stable enough; and the stability bound in the smooth non-convex case is sharper than the existing one. **(2)** We develop a near-optimal high-probability bound of $O(1/\sqrt{n})$ on the transfer error in meta learning. Such bound is also used to reveal the advantage of the S/Q episodic strategy for meta learning over the traditional ERM strategy. **(3)** We derive a deformed generalization bound of $O(\ln n/n)$ with additional curvature condition of loss functions. **(4)** We obtain the first bound for meta learning with dependent episodes. Experiments on regression problems are conducted to validate the convergence of our generalization bounds.

## 2 Related Work

**Algorithmic Stability Theory**. Algorithmic stability analysis is an important tool to provide theoretical guarantee for the learnability of machine learning models. [40] has shown that there are non-trivial problems where traditional uniform convergence analysis (i.e., empirical process theory [44]) fails to hold, but stability can be identified as the sufficient and necessary condition for learnability. There are two main groups in this direction: **(1)** The first group develop different notations of stability and connect their relation to the generalization of specific algorithms. Among them, uniform stability is the most widely used notation and has been utilized to analyze the stability and generalization of regularized ERM algorithms [6]. Hypothesis stability is a weaker notation and has been used to show the stability of $k$-Nearest Neighbor model [12]. Both of the above algorithmic stability notions have been extended to the randomized setting to demonstrate the stability of Bagging algorithm [15]. In recent years, different notations have been employed to analyze the stability and in-expectation generalization bounds of stochastic gradient descent method, which include uniform stability [24], on-average stability [29], uniform argument stability [33, 2], on-average model stability [31] and locally elastic stability [10]. **(2)** The second group aims to derive tight high-probability generalization bounds for uniformly stable algorithm in single-task learning. The first high-probability bound has been derived by [6], and has been improved in [18]. Recently, nearly

optimal generalization bounds of $O(1/\sqrt{n})$ have been established in [19, 7], where $n$ is the size of training dataset. Further, with additional Bernstein condition, [27] derives a generalization bound of $O(1/n)$. In this work, we aim to provide tight stability bounds and improved high-probability generalization bounds for episodic meta learning algorithms. The key step to achieve our goal is to reveal the equivalence of notations between single-task learning and episodic meta learning, hence we can extend the demonstration techniques from [7, 27, 47] to the episodic meta learning setting.

**Generalization Bounds for Meta Learning**. Supposing the $n$ training tasks and the novel tasks are sampled independently from the same environment, [4] derives the first generalization bound on the *transfer error* over the novel task for meta learning. Under the independent task environment assumption, we can categorize existing transfer error bounds into three main groups: **(1)** transfer error bounds of hypothesis space. Such bounds are always achieved via covering number analysis [4] or VC theory [5], and hence are always dimension-dependent. The latest upper bound in this group is of $O(1/\sqrt{nm} + 1/\sqrt{m})$ in [22, Theorem 5], where $m$ is the sample size per task. **(2)** transfer error bounds of the hyper-distribution of prior. Such bounds are obtained via PAC-Bayes analysis [38, 39, 13]. The tightest bound in this group is of order $O(1/\sqrt{n} + 1/m)$ in [17, Theorem 3]. **(3)** transfer error bounds of the algorithm. Such bounds are obtained via algorithmic stability analysis [34, 1]. The tightest bound in this group is of $O(1/\sqrt{n})$ in [8, Theorem 4] for episodic meta learning algorithms. Detailed comparisons between different transfer error bounds can be found in Table A.2 of Appendix A. There also exist other works without the task environment assumption. Instead, they choose to bound the excess risk on the novel task by proposing task-similarity measurement [14, 43], or using the total variation distance as the diversity measurement between novel task and training tasks [16]. In this work, we take the task environment assumption and follow the work of [8]. Our first improvement is to demonstrate that the bound of $O(1/\sqrt{n})$ is near optimal. Besides, we show how to obtain a deformed generalization bound of $O(\ln n/n)$ with additional curvature assumption of the loss function. Further, we derive a bound with dependent training episodes, revealing how dependency relation between episodes can affect the generalization of meta learning algorithms.

## 3 Problem Formulation

In supervised learning, a sample space $\mathcal{Z} = \mathcal{X} \times \mathcal{Y}$ is a product space of an input space $\mathcal{X}$ and an output space $\mathcal{Y}$. $\mathcal{H} = \{h_w : w \in \mathcal{W}\}$ is the hypothesis space where the hypothesis $h_w \in \mathcal{H}$ is parameterized by parameter $w$ in the parameter space $\mathcal{W}$. A measurable function $f : \mathcal{H} \times \mathcal{Z} \to [0, M](M > 0)$ is defined as a nonnegative and bounded loss function, whose loss of a hypothesis $h_w$ over a sample $z$ is denoted by $f(h_w, z)$ or $f(w, z)$. Let $\mathcal{M}_1(A)$ denote the set of probability measures over the set $A$.

**Loss Functions**. Throughout we assume that the parameter space $\mathcal{W} \subset \mathbb{R}^d$. Thus, we use unambiguously $||\cdot|| = ||\cdot||_2$ as the Euclidean norm. Let $\mathrm{Proj}_{\mathcal{W}}$ be the Euclidean projection onto $\mathcal{W}$, which is nonexpansive $||\mathrm{Proj}_{\mathcal{W}}(u) - \mathrm{Proj}_{\mathcal{W}}(v)|| \le ||u - v||$. For any fixed $z \in \mathcal{Z}$, a function $f(\cdot, z) : \mathcal{W} \to \mathbb{R}$ is convex if for all $u, v \in \mathcal{W}$, $f(u, z) \ge f(v, z) + \langle g, u - v \rangle$, where $g \in \partial f(v, z)$, and $\partial f(v, z)$ denotes the set of subgradients of $f(\cdot, z)$ at $v$. Let $\partial^0 f(v, z)$ denote the subgradient with the least norm. If $f(\cdot, z)$ is differentiable, $\partial f(\cdot, z)$ denotes the gradient of $f(\cdot, z)$, i.e., $\partial f(\cdot, z) = \{\nabla f(\cdot, z)\}$. For any $z \in \mathcal{Z}$, $f(\cdot, z)$ is $\sigma$-Lipschitz if $\forall u, v \in \mathcal{W}$, $|f(u, z) - f(v, z)| \le \sigma ||u - v||$. For any $z \in \mathcal{Z}$, $f(\cdot, z)$ is $G$-smooth if $\forall u, v \in \mathcal{W}$, $||\partial f(u, z) - \partial f(v, z)|| \le G||u - v||$. We also give the definition of function with $(\alpha, G)$-Hölder continuous subgradient as follows. We may refer to such functions as $(\alpha, G)$-Hölder smooth or $\alpha$-Hölder smooth function for simplicity when the context is clear.

**Definition 1** *Let $G > 0$, $\alpha \in [0, 1]$. For any $z \in \mathcal{Z}$, a function $f(\cdot, z)$ is called $(\alpha, G)$-Hölder smooth if its subgradient is $(\alpha, G)$-Hölder continuous, i.e., $\partial f(\cdot, z)$ satisfies the following conditions:*

$$\forall u, v \in \mathcal{W}, \quad ||\partial f(u, z) - \partial f(v, z)|| \le G||u - v||^\alpha. \tag{1}$$

If (1) holds with $\alpha = 1$, then $f(\cdot, z)$ is a $G$-smooth function; if (1) holds with $\alpha = 0$, this implies the subgradient boundedness of $f(\cdot, z)$. The examples of loss functions in machine learning satisfying (1) include the $q$-norm hinge-loss $f(w, z) = \big(\max(0, 1 - y\langle w, x \rangle)\big)^q$ for classification and the $q$-th power absolute distance loss $f(w, z) = |y - \langle w, x \rangle|^q$ for regression, whose subgradients are both $(q - 1, C)$-Hölder continuous for some $C > 0$ if $q \in [1, 2]$ (see [9]). For $(\alpha, G)$-Hölder smooth function, define $c_\alpha = (1 + 1/\alpha)^{\frac{\alpha}{1+\alpha}} G^{\frac{1}{1+\alpha}}$ if $\alpha \in (0, 1]$; and $c_\alpha = \sup_z ||\partial f(0, z)|| + G$, if $\alpha = 0$.

**Single-Task Learning**. The training dataset $S = \{z_j = (x_j, y_j)\}_{j=1}^m$ is given by $m$ independent draws from an unknown distribution $D$ on $\mathcal{Z}$ (i.e., $D \in \mathcal{M}_1(\mathcal{Z})$). An algorithm $A$ takes $S$ as input

and outputs a hypothesis $A(S)$ in $\mathcal{H}$. The set of such algorithms depends only on $\mathcal{H}$ and $\mathcal{Z}$ and will be denoted by $\mathcal{A}(\mathcal{H}, \mathcal{Z})$. In single-task learning, a hypothesis $A(S)$ is always obtained by minimizing the empirical error on $S$: $\hat{L}(A(S), S) \triangleq \frac{1}{m} \sum_{j=1}^{m} f(A(S), z_j)$. The performance of the returned hypothesis $A(S)$ is measured by the expected/generalization error with respect to (w.r.t.) the data distribution $D$: $L(A(S), D) \triangleq \mathbb{E}_{z \sim D} f(A(S), z)$. The goal of learning theory is thus to give a (lower or upper) bound on the expected error based on the empirical error on the training dataset $S$.

**Meta Learning**. Following existing theoretical works for meta learning[4, 34, 38, 8], we assume that the distributions $\{D_i\}_{i=1}^{n}$ associated with different training tasks are drawn from the same task *environment* $\tau$, which is a probability distribution over the set of all data distributions on $\mathcal{Z}$ (i.e., $\tau \in \mathcal{M}_1(\mathcal{M}_1(\mathcal{Z}))$). During the training process, a meta-sample $\mathbf{S} = \{S_i = S_i^{tr} \cup S_i^{ts}\}_{i=1}^{n}$ is available, where $S_i^{tr} \overset{\text{i.i.d.}}{\sim} D_i^K$ of size $K$ is the training set, and $S_i^{ts} \overset{\text{i.i.d.}}{\sim} D_i^q$ of size $q$ is the test set of the $i$-th training task. In this work, we assume that $K + q = m$ for notation convenience. The training set and the test set are also called support set and query set [8], respectively. Let $\mathcal{A}(\mathcal{A}(\mathcal{H}, \mathcal{Z}), \mathcal{Z}^m)$ be the set of meta learning algorithms. For any $\mathbf{A} \in \mathcal{A}(\mathcal{A}(\mathcal{H}, \mathcal{Z}), \mathcal{Z}^m)$, it takes the meta-sample $\mathbf{S} = \{S_i\}_{i=1}^{n}$ as input and outputs an algorithm (inner-task algorithm) $\mathbf{A}(\mathbf{S}) : \cup_{m=1}^{\infty} \mathcal{Z}^m \to \mathcal{H}$. The performance of the learned inner-task algorithm is measured by the expectation of the generalization error w.r.t. the task environment $\tau$, which is defined as the *transfer error* by [34, 8] as follows:

$$er(\mathbf{A}(\mathbf{S}), \tau) \triangleq \mathbb{E}_{D \sim \tau} \mathbb{E}_{S^{tr} \sim D^K} \mathbb{E}_{z \sim D} f(\mathbf{A}(\mathbf{S})(S^{tr}), z). \tag{2}$$

Actually, the environment $\tau$ can define an induced distribution $\mathbf{D}_\tau \in \mathcal{M}_1(\mathcal{Z}^m)$, by setting $\mathbf{D}_\tau(F) = \mathbb{E}_{D \sim \tau} D^m(F)$ for any measurable set $F \subseteq \mathcal{Z}^m$. Define the estimator $\mathbf{l}(\mathbf{A}(\mathbf{S}), S) \triangleq \hat{L}(\mathbf{A}(\mathbf{S})(S^{tr}), S^{ts})$, where $S = S^{tr} \cup S^{ts}$, $S \overset{\text{i.i.d.}}{\sim} D^m$. Then we can rewrite the transfer error as a simple form: $er(\mathbf{A}(\mathbf{S}), \tau) = \mathbb{E}_{S \sim \mathbf{D}_\tau} \mathbf{l}(\mathbf{A}(\mathbf{S}), S)$. This means that, the training error $\mathbf{l}(\mathbf{A}(\mathbf{S}), S)$ is the unbiased version of the transfer error $er(\mathbf{A}(\mathbf{S}), \tau) = \mathbb{E}_{S \sim \mathbf{D}_\tau} \mathbf{l}(\mathbf{A}(\mathbf{S}), S)$. This is similar to the fact that, in single-task learning, the empirical error $f(A(S), z)$ is the unbiased version of the generalization error $L(A(S), D) = \mathbb{E}_{z \sim D} f(A(S), z)$. Therefore, a transfer error bound is formally equivalent to a single-task generalization error bound under the identifications $\mathcal{Z} \leftrightarrow \mathcal{Z}^m$, $f \leftrightarrow \mathbf{l}$, $A \leftrightarrow \mathbf{A}$. The relation of the notations between single-task learning and meta learning is listed in Table B.1 in Appendix B. In practice, it is difficult to minimize $er(\mathbf{A}(\mathbf{S}), \tau)$ directly as we have no information of the environment distribution $\tau$. Instead, we choose to minimize the following empirical risk based on the S/Q episodic training strategy. The goal of meta learning theory is thus to give a bound on the transfer error, based on the *empirical multi-task error* on the meta-sample $\mathbf{S}$:

$$\hat{er}(\mathbf{A}(\mathbf{S}), \mathbf{S}) \triangleq \frac{1}{n} \sum_{i=1}^{n} \hat{L}(\mathbf{A}(\mathbf{S})(S_i^{tr}), S_i^{ts}) = \frac{1}{n} \sum_{i=1}^{n} \mathbf{l}(\mathbf{A}(\mathbf{S}), S_i). \tag{3}$$

**Uniform Stability of Meta Learning Algorithms**. We say two meta-samples $\mathbf{S} = \{S_i\}_{i=1}^{n}$ and $\mathbf{S}' = \{S_i'\}_{i=1}^{n}$ are neighboring, denoted by $\mathbf{S} \simeq \mathbf{S}'$, if they only differ on a single entry, i.e., there exists $i \in [n]$ s.t. $\forall j \neq i, S_j = S_j'$; and $S_i \neq S_i'$. We also define $\mathbf{S}^i = \{S_1, .., S_i', ...S_n\}$ as the neighboring meta sample of $\mathbf{S}$ that differs only on the $i$-th entry. We next define the uniform stability of meta algorithms with episodic training strategy, which is formulated explicitly in [8, Definition 3].

**Definition 2** *(Uniform stability of modern meta learning algorithms) A meta algorithm $\mathbf{A}$ has uniform stability w.r.t. the loss function $\hat{L}$ if the following holds for any meta-sample $\mathbf{S}$ and for any $i \in [n]$, any $D \sim \tau$, $S^{tr} \sim D^K$, $S^{ts} \sim D^q$: $|\hat{L}(\mathbf{A}(\mathbf{S})(S^{tr}), S^{ts}) - \hat{L}(\mathbf{A}(\mathbf{S}^i)(S^{tr}), S^{ts})| \leq \gamma$.*

Since $\mathbf{l}(\mathbf{A}(\mathbf{S}), S) = \hat{L}(\mathbf{A}(\mathbf{S})(S^{tr}), S^{ts})$, we can also define the uniform stability of $\mathbf{A}$ as: $\forall S \sim \mathbf{D}_\tau, \forall i \in [n], |\mathbf{l}(\mathbf{A}(\mathbf{S}), S) - \mathbf{l}(\mathbf{A}(\mathbf{S}^i), S)| \leq \gamma$. Such definition is analogous to the uniform stability of an inner-task algorithm $A$ in single-task learning (see Definition D.1 in Appendix D) under the identifications: $\mathbf{l} \leftrightarrow f, \mathbf{A} \leftrightarrow A, \mathbf{S} \leftrightarrow S$. Thus, we can directly apply the existing uniform stability based generalization bound from single-task learning [6, Theorem 12] to obtain the uniform stability based transfer bound for episodic meta learning [8, Theorem 2], without lengthy and somewhat duplicate proof in [8]. We list such fundamental uniform stability based transfer error bound in Theorem 1 for later comparison. To derive sharper transfer error bounds, our **key step** is to utilize the equivalent relation between the notations of single-task learning and episodic meta learning, thus extending fast-rate bounds in single-task learning [7, 27, 47] to the episodic meta learning setting.

**Theorem 1** *Suppose the S/Q episodic meta learning algorithm* $\mathbf{A}$ *has uniform stability* $\gamma$ *w.r.t. the estimator* $l(\cdot, S)$ *bounded by* $M$. *Then, for any task distribution* $\tau \in \mathcal{M}_1(\mathcal{M}_1(\mathcal{Z}))$, *any* $\delta \in (0, 1)$, *the following inequality holds with probability at least* $1 - \delta$ *over the draw of meta sample* $\mathbf{S}$:

$$er(\mathbf{A}(\mathbf{S}), \tau) \leq \hat{er}(\mathbf{A}(\mathbf{S}), S) + \gamma + (2n\gamma + M)\sqrt{\frac{\ln(1/\delta)}{2n}}.$$

## 4 Uniform Argument Stability Bounds of Meta Learning Algorithms

For a modern meta learning algorithm with deep neural networks [20, 41], we always employ stochastic gradient descent (SGD) method to minimize the empirical error $\frac{1}{n}\sum_{i=1}^{n}\hat{L}(\mathbf{A}(\mathbf{S})(S_i^{tr}), S_i^{ts})$ to learn a good feature embedding. Formally, we define the sampling-with-replacement gradient update rule at $(t + 1)$-th step as: $w_{t+1} = \text{Proj}_{\mathcal{W}}[w_t - \eta_t \partial_{w_t}\hat{L}(\mathbf{A}(\mathbf{S})(S_{i_t}^{tr}), S_{i_t}^{ts})]$, where $i_t$ is independently and identically drawn (i.i.d.) from the uniform distribution $Unif([n])$. Therefore, although $\hat{L}(\mathbf{A}(\mathbf{S})(S_{i_t}^{tr}), S_{i_t}^{ts})$ is the loss only calculated over the query samples $S_{i_t}^{ts}$, it is still related to the support samples $S_{i_t}^{tr}$, and the updated parameter $w_{t+1}$ is also related to $S_{i_t}^{tr}$. Therefore, we define an equivalent empirical loss $\hat{R}(\mathbf{A}(\mathbf{S})(S), S) \triangleq \hat{L}(\mathbf{A}(\mathbf{S})(S^{tr}), S^{ts})$ to indicate that: the empirical loss over the episode $S = S^{tr} \cup S^{ts}$ is related to the whole episode sample $S$, and so is the output hypothesis $\mathbf{A}(\mathbf{S})(S)$. Therefore, for the empirical error $\frac{1}{n}\sum_{i=1}^{n}\hat{R}(\mathbf{A}(\mathbf{S})(S_i), S_i)$, the episode-level SGD update rule is: $w_{t+1} = \text{Proj}_{\mathcal{W}}[w_t - \eta_t \partial_{w_t} R(w_t, S_{i_t})]$. The pseudo code as well as several examples of modern meta learning algorithms can be found in Subsection 4.1. In this section, we provide lower and upper stability bounds for meta learning with sampling-with-replacement SGD method. First, we give the definition of uniform argument stability of episodic meta learning algorithms.

**Definition 3** *(Uniform argument stability of meta learning algorithms). Given a meta learning algorithm* $\mathbf{A}$, *any neighboring meta samples* $\mathbf{S}, \mathbf{S}'$, *and any training episode* $S \in \mathcal{Z}^m$, *we define the uniform argument stability random variable of* $\mathbf{A}$ *as* $\delta_{\mathbf{A}}(\mathbf{S}, \mathbf{S}'; S) = \|\mathbf{A}(\mathbf{S})(S) - \mathbf{A}(\mathbf{S}')(S)\|$.

$\mathbf{A}$ is defined as a uniform argument $\beta$-stable meta learning algorithm if for some $\beta > 0$, we have $\sup_{\mathbf{S} \simeq \mathbf{S}', S} \delta_{\mathbf{A}}(\mathbf{S}, \mathbf{S}'; S) \leq \beta$ or $\sup_{\mathbf{S} \simeq \mathbf{S}', S} \mathbb{E}_{\mathbf{A}} \delta_{\mathbf{A}}(\mathbf{S}, \mathbf{S}'; S) \leq \beta$, where $\mathbb{E}_{\mathbf{A}}$ denote the expectation w.r.t. the internal randomness of $\mathbf{A}$. For a meta learning algorithm with SGD method, the internal randomness of $\mathbf{A}$ comes from the randomness of sampling at each iteration. Note that if $\hat{R}(\cdot, S)$ is a Lipschitz function for any $S \in \mathcal{Z}^m$, the uniform argument stability of $\mathbf{A}$ implies the uniform stability of $\mathbf{A}$ in Definition 2. In this work, we investigate the stability of modern meta learning algorithms with sampling-with-replacement SGD training strategy. Therefore, we will derive lower and upper bounds on $\mathbb{E}_{\mathbf{A}}\|\mathbf{A}(\mathbf{S})(S) - \mathbf{A}(\mathbf{S}')(S)\|$ across different settings in Subsections 4.2-4.3.

### 4.1 Pseudo Code of Modern Meta Learning Algorithms

---

**Algorithm 1** Support/Query Episodic Training based Meta Learning Algorithm

---

1: **Input:** training dataset $\mathbf{S} = \{S_i\}_{i=1}^{n}$ with $S_i = \{S_i^{tr}, S_i^{ts}\}$, # of iterations $T$, learning rates $\eta_t$ ($t \in [T]$).
2: **Initialize:** the parameters of deep neural networks $w_1$.
3: **for** $t = 1$ to $T$ **do**
4:     Uniformly sample one of $n$ training episodes with replacement. Let $i_t$ be the episode index.
5:     $w_{t+1} = \text{Proj}_{\mathcal{W}}\left(w_t - \eta_t \partial \hat{R}(w_t, S_{i_t})\right)$         // episode-level SGD update
6: **end for**
7: **return** $w_{T+1}$

---

We provide several specific meta learning algorithms to illustrate the calculation of loss $\hat{R}(w_t, S_{i_t})$ on the episode $S_{i_t}$ at the $t$-th iteration, where $w_t$ is always the parameters of the feature extractor that is shared across different episodes. For the metric-learning based algorithms [41, 45] in classification, $h_{w_t}$ is regarded as the feature extractor to output the feature vector $h_{w_t}(x)$ with data $x$ as input. Then

$$\hat{R}(w_t, S_{i_t}) = \frac{1}{q}\sum_{(x,y) \in S_{i_t}^{ts}} -\log \frac{\exp\{-\mathrm{d}(h_{w_t}(x), c_y)\}}{\sum_k \exp\{-\mathrm{d}(h_{w_t}(x), c_k)\}},$$

where $c_k = \frac{1}{Norm} \sum_{(x,y) \in S_{i_t}^{tr}, y=k} h_{w_t}(x)$ is the prototype (i.e., averaged vector) of the sample features in the support set $S_{i_t}^{tr}$ with the same class label $k$; $\mathrm{d}(\cdot, \cdot)$ is the distance between two feature vectors, e.g. the Euclidean distance in ProtoNet [41], and the Cosine distance in MatchingNet [45]. For the classifier-learning based meta learning algorithm MetaOptNet [30],

$$\hat{R}(w_t, S_{i_t}) = \frac{1}{q} \sum_{(x,y) \in S_{i_t}^{ts}} - \log \frac{\exp\{\lambda \langle h_{w_t}(x), \phi_y \rangle\}}{\sum_k \exp\{\lambda \langle h_{w_t}(x), \phi_k \rangle\}},$$

where $\{\phi_k\}_{k=1}^K$ are the parameters of the classifier returned by supervised learning algorithms (e.g. SVM) on the support set $S_{i_t}^{tr}$, $\langle, \rangle$ represents the inner product. For the gradient-learning based meta algorithm MAML [20], let $\alpha_t$ be the learning rate of the inner-task algorithm at the $t$-th iteration, then

$$\hat{R}(w_t, S_{i_t}) = \frac{1}{q} \sum_{z \in S_{i_t}^{ts}} f\left(w_t - \frac{\alpha_t}{K} \sum_{z' \in S_{i_t}^{tr}} \partial f(w_t, z'), z\right).$$

### 4.2 Stability Bounds for Nonsmooth Functions with $\alpha$-Hölder Continuous Subgradients

In this subsection, we provide lower and upper stability bounds for episodic meta learning algorithm whose loss function is nonsmooth convex and has $\alpha$-Hölder continuous subgradient with $0 \leq \alpha < 1$.

**Theorem 2** $\forall$ *fixed* $S \in \mathcal{Z}^m$, *let* $\hat{R}(\cdot, S)$ *be a convex and* $(\alpha, G)$-*Hölder smooth function, where* $\alpha \in [0, 1)$. *Let* $\mathbf{A}$ *be a meta learning algorithm with sampling-with-replacement SGD. Denote by* $w_j$ *and* $w_j'$ *the outputs after* $j (j \in [T])$ *steps of SGD on* $\mathbf{S}$ *and* $\mathbf{S}^i$, *respectively. Define* $R_{\mathbf{S}}(w) = n^{-1} \sum_{i=1}^n \hat{R}(w, S_i)$, $\forall w \in \mathcal{W}$. *Then* $\forall S \in \mathcal{Z}^m$, $\mathbb{E}_{\mathbf{A}} \delta_{\mathbf{A}}(\mathbf{S}, \mathbf{S}'; S)$ *is upper bounded by*

$$\sqrt{2} c_\alpha \left[ \sum_{j=1}^T \eta_j^2 \mathbb{E}\left[ R_{\mathbf{S}}^{\frac{2\alpha}{1+\alpha}}(w_j) + R_{\mathbf{S}^i}^{\frac{2\alpha}{1+\alpha}}(w_j') \right] \right]^{\frac{1}{2}} + \frac{2 c_\alpha}{n} \sum_{j=1}^T \eta_j \left[ \hat{R}^{\frac{\alpha}{1+\alpha}}(w_j, S_i) + \hat{R}^{\frac{\alpha}{1+\alpha}}(w_j, S_i') \right]. \tag{4}$$

*In addition, if* $\hat{R}(\cdot, S)$ *is bounded by* $M$ *and the step size* $\eta_j = \eta \; \forall j \in [T]$, *we can obtain the lower and upper bounds of the uniform argument stability of* $\mathbf{A}$: $c_\alpha M^{\frac{\alpha}{1+\alpha}} \left( \min\{1, \frac{T}{n}\} \eta \sqrt{T} + \frac{\eta T}{n} \right) \leq \sup_{\mathbf{S}, \mathbf{S}', S} \mathbb{E}_{\mathbf{A}} \delta_{\mathbf{A}}(\mathbf{S}, \mathbf{S}'; S) \leq 4 c_\alpha M^{\frac{\alpha}{1+\alpha}} \left( \min\{1, \frac{T}{n}\} \eta \sqrt{T} + \frac{\eta T}{n} \right)$.

**Remark 1** *Our upper stability bound in Eq. (4) depends on the empirical risk during the optimization process. Formally, Eq. (4) shows that, the stability of modern meta algorithm increases if we find good parameters* $w_j$ *with small empirical risk* $R_{\mathbf{S}}(w_j)$ *at the* $j$-*th optimization step. This illustrates a key insight that optimization is beneficial to improve the generalization of algorithms. Besides, our stability upper bound also implies the importance of a good embedding [42] (which may have a good initialization and low empirical risk during the first several optimization steps) to generalization.*

**Remark 2** *We additionally suppose the function to be bounded by* $M$ *such that the stability bounds can be used to analyze the generalization bounds in the next section where the loss function is always assumed to be bounded. Note that when* $\alpha = 0$, $\hat{R}(\cdot, S)$ *is a nonsmooth* $c_\alpha$-*Lipschitz convex function, and our lower and upper stability bounds recover the results in [2]. For bounded convex* $\alpha$-*Hölder smooth functions, the lower stability bound implies that modern meta learning algorithms are not stable enough even if we train them with a relatively small constant step size in each SGD iteration.*

**Remark 3** *The above result shows that, for bounded convex* $\alpha$-*Hölder smooth function* ($\alpha \in [0, 1)$), *the uniform argument stability parameter* $\beta = O\left( c_\alpha M^{\frac{\alpha}{1+\alpha}} (\min\{1, T/n\} \eta \sqrt{T} + \eta T/n) \right)$. *Another work [31] also focuses on convex* $\alpha$-*Hölder smooth function. Using the technique from [31], we obtain the upper stability bounds for bounded convex* $\alpha$-*Hölder smooth function under the same conditions:* $\beta \leq O\left( c_\alpha M^{\frac{\alpha}{1+\alpha}} (\eta^{\frac{1}{1-\alpha}} T + \eta T/n) \right)$ *or* $\beta \leq O\left( c_\alpha M^{\frac{\alpha}{1+\alpha}} (\eta^{\frac{1}{1-\alpha}} \sqrt{T} + \eta \sqrt{T/n}) \right)$ *(see Theorems C.1-C.2 in Appendix C.2.2), both of which are larger than our tight stability bound in Theorem 2 under the setting* $T \leq n$. *When* $T > n$, *the upper bound* $\beta \leq O\left( c_\alpha M^{\frac{\alpha}{1+\alpha}} (\eta^{\frac{1}{1-\alpha}} \sqrt{T} + \eta \sqrt{T/n}) \right)$ *in Theorem C.2 is slightly sharper than the upper bound* $O\left( c_\alpha M^{\frac{\alpha}{1+\alpha}} (\eta \sqrt{T} + \eta T/n) \right)$ *in Theorem 2.*

### 4.3 Stability Bounds for Smooth Functions

In this subsection, we provide lower and upper uniform argument stability bounds for modern meta learning algorithms with smooth functions. First, we consider smooth convex functions.

**Theorem 3** $\forall$ *fixed* $S \in \mathcal{Z}^m$, *let* $\hat{R}(\cdot, S)$ *be a G-smooth convex function. Let* $\mathbf{A}$ *be a meta learning algorithm with sampling-with-replacement SGD. Denote by* $w_j$ *and* $w'_j$ *the outputs after* $j(j \in [T])$ *steps of SGD on neighboring meta samples* $\mathbf{S}$ *and* $\mathbf{S}^i$, *respectively. Then* $\forall S \in \mathcal{Z}^m$, $\eta_j \leq 2/G$,

$$\mathbb{E}_{\mathbf{A}} \|\mathbf{A}(\mathbf{S})(S) - \mathbf{A}(\mathbf{S}^i)(S)\| \leq \frac{\sqrt{2G}}{n} \sum_{j=1}^{T} \eta_j \mathbb{E}_{\mathbf{A}} \left[ \sqrt{\hat{R}(w_j, S_i)} + \sqrt{\hat{R}(w'_j, S'_i)} \right].$$

*In addition, if* $\hat{R}(\cdot, S)$ *is bounded by* $M$, *we can obtain the lower and upper bounds of the uniform argument stability of* $\mathbf{A}$: $\frac{1}{n} \sum_{j=1}^{T} \eta_j \leq \sup_{\mathbf{S},\mathbf{S}',S} \mathbb{E}_{\mathbf{A}} \delta_{\mathbf{A}}(\mathbf{S}, \mathbf{S}'; S) \leq \frac{2\sqrt{2MG}}{n} \sum_{j=1}^{T} \eta_j$.

If we set all $\eta_j = \eta$, then for bounded convex functions, the tight stability bound of $O(\frac{\eta T}{n})$ under the smooth case is sharper than the stability bound of $O(\min\{1, \frac{T}{n}\}\eta\sqrt{T} + \frac{\eta T}{n})$ in Theorem 2 under the nonsmooth case. This indicates that in the smooth case, meta learning algorithms are more stable than in the nonsmooth case. Finally, we give stability bounds for smooth non-convex functions.

**Theorem 4** $\forall$ *fixed* $S \in \mathcal{Z}^m$, *let* $\hat{R}(\cdot, S)$ *be a* $\sigma$-*Lipschitz and G-smooth function. Let* $\mathbf{A}$ *be a meta learning algorithm. Denote by* $w_j$ *and* $w'_j$ *the outputs after* $j(j \in [T])$ *steps of SGD on* $\mathbf{S}$ *and* $\mathbf{S}^i$, *respectively. Define the learning rate* $\eta_j = \frac{a}{jG}$, $\forall j \in [T]$ *with* $a > 0$. *Then* $\forall S \in \mathcal{Z}^m$, *the lower and upper stability bounds of* $\mathbf{A}$ *satisfy:* $\frac{T^a}{6n^{1+a}} \leq \sup_{\mathbf{S},\mathbf{S}',S} \mathbb{E}_{\mathbf{A}} \delta_{\mathbf{A}}(\mathbf{S}, \mathbf{S}'; S) \leq \frac{11 \ln(n)\sigma T^a}{n^{1+a}}$.

Under the same step size setting, existing upper uniform argument stability bound in [8, Theorem3] or in [33, Proposition 4] for non-convex, smooth and Lipschitz function is of $O(T^{\frac{a}{a+1}}/n)$. Our bound of order $O(T^a/n^{1+a})$ is improved over the existing bound when $T^{\frac{a}{1+a}} \leq n$. Besides, our stability bound can be non-vacuous for multi-pass SGD setting (i.e., when $T = kn$, $k \in \mathbb{N}$) where the number $T$ of SGD iterations is larger than $n$, as long as $k \leq n^{1/a}$.

## 5 High Probability Transfer Error Bounds for Meta Learning

In this section, we establish high probability bounds for transfer error $er(\mathbf{A}(\mathbf{S}), \tau)$. Specifically, we still consider two kinds of loss function: **(1)** convex and $(\alpha, G)$-Hölder smooth function ($\alpha \in [0, 1]$); **(2)** non-convex, $\sigma$-Lipschitz and $G$-smooth function. We always assume that the loss function $\hat{R}(\cdot, \cdot)$ is bounded by $M$. Define $\sigma_\alpha = c_\alpha M^{\frac{\alpha}{1+\alpha}}$ if $\hat{R}(w, S)$ is a convex and $(\alpha, G)$-Hölder smooth function; $\sigma_\alpha = \sigma$ if $\hat{R}(w, S)$ is a $\sigma$-Lipschitz and $G$-smooth function. We just exhibit the generalization bounds of randomized algorithm $\mathbf{A}$ by supposing $\mathbb{P}_{\mathbf{A}}[\delta_{\mathbf{A}}(\mathbf{S}, \mathbf{S}'; S) > \beta] \leq \delta_0$. We provide an example to illustrate how to calculate $\delta_0$ in Example D.1 in Appendix D. The generalization bound for deterministic meta algorithm (e.g. with gradient descent) can be stated by setting $\delta_0 = 0$.

### 5.1 Near Optimal Transfer Error Bound for Meta Learning with Independent Episodes

We denote by $a \lesssim b$ the existence of some universal constant $c > 0$ such that $a \leq cb$. Then we obtain the following near optimal bound of $O(1/\sqrt{n})$ under the independent task environment assumption.

**Theorem 5** *Let* $\mathbf{A} \in \mathcal{A}(\mathcal{A}(\mathcal{H}, \mathcal{Z}), \mathcal{Z}^m)$ *be a uniform argument* $\beta$-*stable meta algorithm, i.e.,* $\sup_{\mathbf{S} \simeq \mathbf{S}',S} \mathbb{E}_{\mathbf{A}} \|\mathbf{A}(\mathbf{S})(S) - \mathbf{A}(\mathbf{S}')(S)\| \leq \beta$. *For any* $S \in \mathcal{Z}^m$, *let* $\hat{R}(\cdot, S)$ *be* $[0, M]$-*valued, and satisfy one of the two following conditions: (1)* $\hat{R}(\cdot, S)$ *is convex and* $(\alpha, G)$-*Hölder smooth* $(\alpha \in [0, 1])$; *(2)* $\hat{R}(\cdot, S)$ *is* $\sigma$-*Lipschitz and G-smooth. Suppose* $\mathbb{P}_{\mathbf{A}}[\delta_{\mathbf{A}}(\mathbf{S}, \mathbf{S}'; S) > \beta] \leq \delta_0$. *Then for any independent task environment* $\tau \in \mathcal{M}_1(\mathcal{M}_1(\mathcal{Z}))$, *any* $\delta \in (0, 1)$, *the following holds with probability at least* $1 - \delta - \delta_0$ *over the draw of* $\mathbf{S}$ *and the internal randomness of* $\mathbf{A}$:

$$\sigma_\alpha \beta \ln \frac{1}{\delta} + \frac{M}{\sqrt{n}} \sqrt{\ln(1/\delta)} \lesssim er(\mathbf{A}(\mathbf{S}), \tau) - \hat{er}(\mathbf{A}(\mathbf{S}), S) \lesssim \sigma_\alpha \beta \ln \frac{n}{\delta} + \frac{M}{\sqrt{n}} \sqrt{\ln(1/\delta)}.$$

**Remark 4** *Our transfer error bound in Theorem 5 has three advantages over the bound in Theorem 1 from [8]: (1) Theorem 1 gives a high-probability upper bound of* $O(\sqrt{n}\gamma + M/\sqrt{n})$ *for transfer error, where* $\gamma$ *is the uniform stability parameter and always scales as* $O(1/n)$; *in contrast, our upper bound of* $O(\beta \ln n + M/\sqrt{n})$ *is improved by replacing the* $\sqrt{n}$ *factor before the stability parameter with* $\ln n$. *(2) In [8], the uniform stability* $\gamma = O(T^{\frac{a}{a+1}}/n)$, *whereas our uniform argument stability* $\beta = O(T^a/n^{1+a})$ *is tighter when* $T^{\frac{a}{a+1}} \leq n$, *i.e., when the uniform stability bound* $\gamma = O(T^{\frac{a}{a+1}}/n)$ *is non-vacuous. (3) Our high-probability transfer error bound of order* $O(1/\sqrt{n})$ *is near optimal.*

**Remark 5** *We uncover two limitations of stability-based meta learning theory: (1) Recall the lower stability bound for meta learning algorithms with convex $\alpha$-Hölder smooth function ($\alpha \in [0,1)$) in Theorem 2, we find that the lower transfer error bound in Theorem 5 is $er(\mathbf{A}(\mathbf{S}), \tau) - \hat{er}(\mathbf{A}(\mathbf{S}), \mathbf{S}) \gtrsim \sigma_\alpha \ln(1/\delta) c_\alpha M^{\frac{\alpha}{1+\alpha}} (\eta\sqrt{T} + \eta T/n)$ when $T \geq n$. This indicates that the lower transfer error bound is greater than a constant and will not converge to zero with the increase of $n$. Thus, the stability-based transfer error bound is vacuous and cannot provide asymptotic guarantees for convex Hölder smooth functions. (2) The stability-based transfer error bound of $O(1/\sqrt{n})$ in Theorem 5 is near optimal. Such result is consistent with the observation in [36, Section 2] that under the (i.i.d.) task environment assumption, the term $O(1/\sqrt{n})$ in the generalization bound is unavoidable. Thus, to obtain sharper generalization bounds for meta learning (e.g. the bound of $O(1/\sqrt{mn})$ or even $O(1/mn)$), we need to consider other stability notions (e.g. [16]), or suppose stronger task relatedness in the environment (e.g. [5, 22]), or even drop the task environment assumption (e.g. [14, 43]).*

**Remark 6** *Under the independent task environment assumption, we compare our bound of $O(1/\sqrt{n})$ via S/Q episodic training strategy with other transfer error bounds that are obtained via traditional ERM strategy over all samples in training tasks. In detail, the bound from [34, Theorems 2 and 6] via algorithmic stability analysis is of $O(1/m + 1/\sqrt{n})$; the bounds from [38, Theorem 1] and [39, Theorem 2] via PAC-Bayes analysis are of $O(1/\sqrt{n} + 1/\sqrt{m})$; the bound from [22, Theorem 5] via covering number analysis is of $O(1/\sqrt{nm} + 1/\sqrt{m})$. All of these bounds via ERM strategy involve a term $O(1/\sqrt{m})$, and such term can be large when $m$ is relatively small (e.g. $m = 5$ or $m = 10$ in the few-shot learning setting). Thus, in terms of the tightness of transfer error bounds, the S/Q episodic training strategy is superior to the ERM strategy for meta learning, when $m << n$. Such result was also pointed out by [8] and is more rigorously demonstrated in this work. Detailed comparisons between different transfer error bounds for meta learning are shown in Table A.2 in Appendix A.*

## 5.2 Fast Transfer Error Bound of $O(\ln n/n)$ for Meta Learning with Independent Episodes

To obtain faster convergence rate, we need to take additional assumption. The generalized Bernstein condition is one of the most widely used condition to attain fast convergence rate of generalization bound in single-task learning [33, 27]. Next, we extend the generalized Bernstein condition to the meta learning setting, where we study the optimal algorithm $A^*$ instead of the optimal hypothesis.

**Definition 4** *(Generalized Bernstein Condition for Meta Learning) Assume that $A^*(\mathcal{H}, \mathcal{Z}) = Argmin_{A \in \mathcal{A}(\mathcal{H}, \mathcal{Z})} er(A, \tau)$ is a set of risk minimizers in a closed set. We say that an algorithm $A$ together with the environment $\tau$ and the empirical estimator $\mathbf{l}$ satisfies the generalized Bernstein condition if for some $B > 0$, $\forall A \in \mathcal{A}(\mathcal{H}, \mathcal{Z})$, there is a $A^* \in A^*(\mathcal{H}, \mathcal{Z})$, such that*

$$\mathbb{E}_{S \sim \mathbf{D}_\tau} \big( \mathbf{l}(A, S) - \mathbf{l}(A^*, S) \big)^2 \leq B \big( er(A, \tau) - er(A^*, \tau) \big). \tag{5}$$

[27] has shown that in single-task learning, a strongly-convex and Lipschitz function satisfies the generalized Bernstein condition. In this work, we relax the strong-convexity condition by considering the following Polyak-Łojasiewicz condition, one of the weakest curvature conditions of functions.

**Definition 5** *(Polyak-Łojasiewicz [46]) Any function $f : \mathcal{W} \to \mathbb{R}$ satisfies the Polyak-Łojasiewicz (PL) condition on $\mathcal{W}$ with parameter $\mu > 0$ if for all $w \in \mathcal{W}$, $f(w) - f(w^*) \leq \frac{1}{2\mu}\|\partial^0 f(w)\|_2^2$, where $w^*$ denotes the Euclidean projection of $w$ onto the set of global minimizer of $f$ in $\mathcal{W}$.*

A key insight into the PL condition is that it is the sufficient and necessary condition to guarantee the linear convergence of gradient descent methods for smooth convex optimization problem [37]. Such PL condition can also be satisfied by many non-convex neural network models, such as the two-layer neural networks with ReLU activation functions [32] and the deep linear residual networks [23]. We will show that if the functions in Theorem 5 additionally satisfy the PL condition, then the loss functions in meta learning also satisfy the generalized Bernstein condition in Definition 4. Thus, we can derive a "deformed" transfer error bound of $O(\ln n/n)$ for modern meta learning algorithms.

**Theorem 6** *Under the same conditions of Theorem 5, for any fixed $S \in \mathcal{Z}^m$, let $\hat{R}(\cdot, S)$ additionally satisfy Polyak-Łojasiewicz condition in Definition 5. Suppose $\mathbb{P}_\mathbf{A}[\delta_\mathbf{A}(\mathbf{S}, \mathbf{S}'; S) > \beta] \leq \delta_0$. Then, there exist $c > 0$, such that for any environment $\tau \in \mathcal{M}_1(\mathcal{M}_1(\mathcal{Z}))$, and any $\delta \in (0, 1)$, the following holds with probability at least $1 - \delta - \delta_0$ over the draw of $\mathbf{S}$ and the internal randomness of $\mathbf{A}$:*

$$er(\mathbf{A}(\mathbf{S}), \tau) \leq (1+\eta)\hat{er}(\mathbf{A}(\mathbf{S}), \mathbf{S}) + c(1 + 1/\eta)\Big(\sigma_\alpha \beta \ln n + \frac{M}{n}\Big)\ln\frac{1}{\delta}.$$

Recall in Section 4, our stability parameter is always $\beta = O(1/n)$. Hence, when the empirical error in the RHS of the above bound is close to zero, the transfer error bound always scales as $O(\ln n/n)$.

### 5.3 Transfer Error Bound for Meta Learning with Dependent Episodes

In this subsection, we investigate the generalization bound for meta learning algorithms with dependent episodes whose dependency relation can be characterized by a graph. The approach undertaken to establish our results is based on the forest approximation of the dependency graph [26]. Formally, a dependency graph is an undirected graph $\Gamma = (V, E)$ of a random vector $\mathbf{S} = (S_1, ..., S_n)$ if the following two conditions are satisfied: (1) $V(\Gamma) = [n]$; (2) if $I, J \subset [n]$ are non-adjacent in $\Gamma$, then $\{S_i\}_{i \in I}$ and $\{S_j\}_{j \in J}$ are independent. We next give the concept of forest approximation.

**Definition 6** *(Forest Approximation [47]) Given a graph $\Gamma$, a forest $F$, and a mapping $\phi : V(\Gamma) \to V(F)$, if $\phi(u) = \phi(v)$ or $\langle \phi(u), \phi(v) \rangle \in E(F)$ for any $\langle u, v \rangle \in E(\Gamma)$, we say that $(\phi, F)$ is a forest approximation of $\Gamma$. Let $\Phi(\Gamma)$ denote the set of forest approximations of $\Gamma$.*

Intuitively, a forest approximation transform a graph into a forest by merging vertices and removing self-loops. We then give the definition of forest complexity, which measures how a dependency graph looks like a forest, and hence measures the strength of dependency among random variables in $\mathbf{S}$.

**Definition 7** *(Forest Complexity [47]) Given a graph $\Gamma$ and any forest approximation $(\phi, F) \in \Phi(G)$ with $F$ consisting of trees $\{T_i\}_{i \in [k]}$. Define $\lambda_{(\phi, F)} = \sum_{\langle u,v \rangle \in E(F)} \left( |\phi^{-1}(u)| + |\phi^{-1}(v)| \right)^2 + \sum_{i=1}^{k} \min_{u \in V(T_i)} |\phi^{-1}(u)|^2$. We call $\Lambda(\Gamma) = \min_{(\phi, F) \in \Phi(\Gamma)} \lambda_{(\phi, F)}$ the forest complexity of the graph $\Gamma = (V, E)$. Here, $\phi^{-1}(u)$ is the set of pre-images of the element $u$.*

For sample $\mathbf{S}$ whose components are independent, we choose the identity map and its dependency graph as the forest approximation. Hence $\Lambda(\Gamma) = \sum_{i=1}^{n} 1^2 = n$. For sample $\mathbf{S}$ whose dependency graph $\Gamma$ is a tree, the identity map between $\Gamma$ and itself is a forest approximation of $\Gamma$. Then $\Lambda(\Gamma) \le |E(\Gamma)|(1+1)^2 + 1 = 4n - 3 = O(n)$. More examples of forest approximation can be found in [47, Section 3.3]. We next give a forest-complexity based transfer error bound for meta learning.

**Theorem 7** *Under the same conditions of Theorem 5, except that $\mathbf{S}$ is a meta sample of size $n$ with dependency graph $\Gamma$. Let the maximum degree of the graph $\Gamma$ is $\triangle$. Suppose $\mathbb{P}_{\mathbf{A}}[\delta_{\mathbf{A}}(\mathbf{S}, \mathbf{S}'; S) > \beta] \le \delta_0$. Then, for any environment $\tau \in \mathcal{M}_1(\mathcal{M}_1(\mathcal{Z}))$, any $\delta \in (0, 1)$, the following holds with probability at least $1 - \delta - \delta_0$ over the draw of $\mathbf{S}$ and the internal randomness of $\mathbf{A}$:*

$$er(\mathbf{A}(\mathbf{S}), \tau) \le \hat{er}(\mathbf{A}(\mathbf{S}), \mathbf{S}) + \sigma_\alpha \beta (\triangle + 1) + \left(2\sigma_\alpha \beta + \frac{M}{n}\right) \sqrt{\frac{\Lambda(\Gamma) \ln 1/\delta}{2}},$$

When $\mathbf{S}$ is an independent sample, the forest complexity $\Lambda(\Gamma) = n$, the maximum degree $\triangle = 0$, and the above forest-complexity based generalization bound degenerates to the bound in Theorem 1 for meta learning with independent episodes. When $\mathbf{S}$ is a dependent sample, $\Lambda(\Gamma)$ will be greater than $n$. Both the complexity $\Lambda(\Gamma)$ and the maximum degree $\triangle$ will increase with more dependency relation between samples in $\mathbf{S}$ (i.e., with more adjacent edges in its dependency graph $\Gamma = (V, E)$). In the next section, we conduct experiments on regression problems to show the convergence performance of the generalization bound for meta learning with dependent episodes. The corresponding forest-complexity based generalization bound for such problem is provided in Example D.2 in Appendix D.3.

## 6 Experiments

To verify our theoretical analysis, we conduct experiments on few-shot regression problems to show the convergence performance of our generalization bounds with independent and dependent episodes.

**Experimental Settings**. We follow the experimental setting in [20, 8]. The problem aims to approximate the distribution of parameters of function $f(x) = \alpha \sin(\beta x)$. The task environment $\tau$ is the joint distribution $p(\alpha, \beta)$ of the parameters $\alpha$ and $\beta$. We set $p(\alpha) = U[-5, 5]$, $p(\beta) = U[0, \pi]$. For independent setting, we construct training episodes by sampling pairs $(\alpha, \beta)$ from the task distribution $\tau = p(\alpha, \beta)$; for dependent setting, we construct the first half training episodes by sampling pairs $(\alpha, \beta)$ from $\tau = p(\alpha, \beta)$ independently, and construct the rest half training episodes by setting $(-\alpha, \pi - \beta)$ with $(\alpha, \beta)$ from the first half training episodes. Each training episode contains

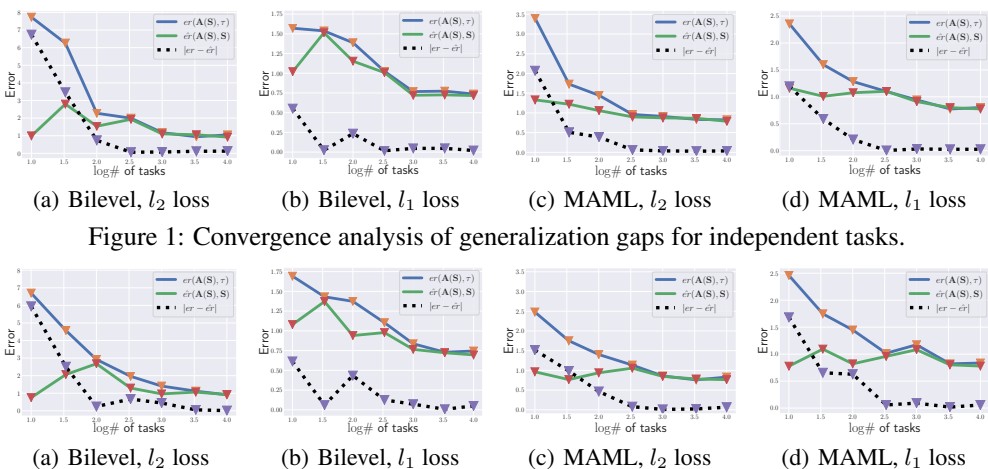

Figure 1: Convergence analysis of generalization gaps for independent tasks.

(a) Bilevel, $l_2$ loss     (b) Bilevel, $l_1$ loss     (c) MAML, $l_2$ loss     (d) MAML, $l_1$ loss

(a) Bilevel, $l_2$ loss     (b) Bilevel, $l_1$ loss     (c) MAML, $l_2$ loss     (d) MAML, $l_1$ loss

Figure 2: Convergence analysis of generalization gaps for dependent tasks.

$5$ support samples and $1$ query samples. In both settings, the $600$ test episodes are constructed by sampling $(\alpha, \beta)$ from $\tau$ independently, each containing $5$ support samples and $15$ query samples. We implement meta learning algorithms MAML [20] and Bilevel Programming [21] with square loss ($l_2$) and absolute loss ($l_1$). The neural network has two hidden layers of size $40$ with ReLU activation functions. The generalization gap is the absolute distance between the training error and test error.

**Experimental Results**. From Figures 1-2, we can observe that: **(1)** The generalization gap in both independent and weakly dependent episode settings can converge to $0$ with the increase of the training episodes, demonstrating the asymptotic behaviour of our transfer error bounds in Theorems 5 and 7. **(2)** The generalization gap with independent episodes can converge to zero more quickly than the gap with dependent episodes. The test error with independent episodes also always converge to the lower level than the one with dependent episodes. The better convergence performance with independent episodes truly demonstrate how the dependency between episodes can affect the generalization of meta learning algorithms. **(3)** With non-convex neural network models, both square loss and nonsmooth absolute loss can lead to similar convergence performance of generalization bounds.

## 7 Conclusion and Future Work

In this work, we provide fine-grained analysis of stability and generalization for modern meta learning algorithms. From the perspective of stability, our tight stability bounds implies that in the nonsmooth convex case the meta learning algorithm is less stable than in the smooth convex case. The stability bounds in the smooth non-convex case enjoys an order of $O(1/n)$ even for the multi-pass SGD setting. From the perspective of generalization, we demonstrate that the high-probability transfer error bound of $O(1/\sqrt{n})$ is optimal. Based on this bound, we uncover the limitations of algorithmic stability analysis for meta learning, and reveal the advantage of episodic training strategy for meta learning over tradition ERM training strategy. Further, by extending the generalized Berstein condition to the meta learning setting, we obtain a deformed generalization bound of $O(\ln n/n)$ with additional Polyak-Łojasiewicz condition. Finally, we derive a generalization bound for meta learning with dependent episodes. Experiments are also provided to show the convergence performance of generalization error with independent and dependent episodes. In the future, we will explore new stability notions to see whether we can develop sharper generalization bounds for meta learning.

## Acknowledgments and Disclosure of Funding

Jiechao would like to thank Dr. Mingxue Quan from School of Mathematics in Renmin University of China for helpful discussions. We thank all reviewers for their constructive suggestions to improve the quality of this paper. This work was supported in part by National Natural Science Foundation of China (61976220 and 61832017), Beijing Outstanding Young Scientist Program (BJJWZYJH012019100020098), and Large-Scale Pre-Training Program 468 of Beijing Academy of Artificial Intelligence (BAAI). Prof. Zhiwu Lu is the corresponding author of this paper.

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
