# OpenReview forum: "Fine-Grained Analysis of Stability and Generalization for Modern Meta Learning Algorithms"
_NeurIPS.cc/2022/Conference — NeurIPS 2022 Accept_

### Official Review · Reviewer_E4JU · 2022-06-27

**Rating:** 7
**Confidence:** 2
**Soundness:** 3 good
**Presentation:** 3 good
**Contribution:** 3 good

**Summary:**

This work provides a comprehensive stability analysis of gradient-based modern meta-learning algorithms, i.e. that use Support/Query (S/Q) splits of the task dataset for meta-training. In particular, the authors provide matching upper and lower stability bounds for  convex loss functions with $(\alpha, G)$-Holder subgradients and for smooth functions. The bounds show that in the non-smooth convex case, meta-learning algorithms are inherently less stable than in the smooth convex case. In particular, the lower bound in the non-smooth case is vacuous. In the smooth case, the bounds are sharper than previous analysis. Furthermore, they also develop a near-optimal generalization upper bound in high probability, which shows the advantage of S/Q training compared to  empirical risk minimization (ERM). With the additional PL curvature condition, the authors derive a faster generalization bound. Finally, they derive  bounds for the case where the  episodes have a dependency relation encoded in a graph. Experiments on a synthetic regression problem with L1 and L2 loss validate the theoretical findings.

**Questions:**

Major question and comments.
1. Theorem 6 does not contain a valid bound since in the RHS it has $(1+ \eta) er$ instead of just $er$. From the analysis we can see that this result follows from [12, Theorem 1.2], which specifies that it is useful only when the empirical error is small, which might happen when dealing with overparameterized neural networks. The correct fast bound is [12, Theorem 1.1] which bounds the excess risk instead of the generalization error. I think this should be clarified and claims should be modified accordingly.
2. The advantage of S/Q meta-learning over ERM has already been shown by [8]. I think readers might mistakenly think that this is a new result of the paper. I suggest removing this fact from the claims in the introduction and possibly cite [8] in Remark 6.
3. Proof of Theorem 5 uses Lemma E.1 in the appendix, which should be equivalent to [5, Corollary 8]. However, such corollary is only an upper bound, the lower bound is derived in the subsequent section of [5]. I think this should be clarified and the proof should be expanded.
4. I suggest the authors either include log terms inside $O()$ or use $\tilde{O}(\cdot)$.


Minor questions and typos:
1. Looking at [6, Theorem 12], I think you should replace $\gamma$ with $2\gamma$ in the bound of Theorem 1.
2. Addiction -> addition.
3. Theorem 1 corresponds to [8, Theorem 2], not [8, Theorem 1]
4. In Remark 6 “m=1 in few-shot learning”. If m=1 is impossible to split the dataset in support and query. At least should be m=2 but in practice is larger than that.

References:

[5]  O. Bousquet, Y. Klochkov, and N. Zhivotovskiy. Sharper bounds for uniformly stable algorithms. In Conference on Learning Theory (COLT), pages 610–626, 2020.

[6] O. Bousquet and A. Elisseeff. Stability and generalization. Journal of Machine Learning Research (JMLR), 2:499–526, 2002.

[8] J. Chen, X. Wu, Y. Li, Q. LI, L. Zhan, and F. Chung. A closer look at the training strategy for modern meta-learning. In Conference on Neural Information Processing Systems (NeurIPS), pages 396–406, 2020.

[12] Y. Klochkov and N. Zhivotovskiy. Stability and deviation optimal risk bounds with convergence rate o(1/n). In Conference on Neural Information Processing Systems (NeurIPS), 2021.





**Limitations:**

The authors adequately discuss the limitations of the analysis in Remark 5.


**Strengths And Weaknesses:**

Strengths:
+ Original and significant results on S/Q meta-learning. In particular, I am not aware of lower bounds for this setting.
+ Clearly written.

Weaknesses:
- Some incorrect and not novel claims. (See Question 1 and 2)
- The case with vacuous lower-bound is not analyzed in the experiments.

**Update after the discussion with the authors**

The authors properly addressed my main concerns by updating the manuscript. Therefore I increase the score from 6 to 7

---

> ### Author Response · Authors · 2022-08-01
> **Response to the Review by Reviewer E4JU**
>
> **Q1. Theorem 6 does not contain a valid bound since in the RHS it has (1+η)er instead of just er**.\
> A: Thanks. It is true that the bound in the RHS in Theorem 6 is actually the so-called “deformed” transfer error bound, where there is a larger-than-1 multiplicative factor in front of the empirical multi-task error $er$. When the empirical multi-task error is close to zero, the transfer error has a convergence rate of $O(\ln{n}/n)$. We have added these discussions below Theorem 6 in the revised version.
>
> **Q2. The advantage of S/Q meta-learning over ERM has already been shown by [8]. I suggest removing this fact from the claims in the introduction and possibly cite [8] in Remark 6**. \
> A: Thanks for your suggestions. Although [8] has shown the advantage of S/Q meta-learning, our work makes improvements in two aspects: \
> $(1)$ We conduct a more comprehensive comparison between our bound and other transfer error bounds obtained via traditional ERM strategy, to show the advantage of S/Q meta-learning. Concretely, in Remark 6, we compare our stability-based bound for S/Q meta-learning with transfer error bounds obtained with PAC-Bayes analysis [37,38],  the bounds with model-capacity theory [4,22],  and the bounds with algorithmic stability analysis [33]; in contrast, [8] only compared their bound for S/Q meta-learning with the stability-based bound for traditional meta-learning in [33]. \
> $(2)$ Our comparisons are more accurate: directly comparing different generalization UPPER bounds, to some extent, is not so accurate. However, our work has shown that the transfer error bound of $O(1/\sqrt{n})$ is near optimal, and compared such bound with others to show the advantage of S/Q meta-learning. Therefore, our comparisons are more accurate. Nevertheless, we have refined our statements in the introduction and in Remark 6 to clarify our contributions in the revised version.
>
> **Q3. Proof of Theorem 5 uses Lemma E.1 in the appendix, which should be equivalent to [5, Corollary 8]. However, the lower bound is derived in the subsequent section of [5]**.\
> A: Thanks. We have clarified the citation of Lemma E.1 in the revised version.
>
> **Q4. I suggest the authors either include log terms inside $O()$**.\
> A: Thanks for your suggestion. We have included log terms inside $O(\cdot)$ throughout the paper in the revised version.
>
> **Q5. Looking at [6, Theorem 12], I think you should replace $\gamma$ with $2\gamma$ in the bound of Theorem 1**.\
> A: Thanks for pointing this out. Our Theorem 1 is true. The reason for the difference between our Theorem 1 and [6, Theorem 12] is that the uniform stability notion in our Theorem 1 is slightly different from the uniform stability notion in [6]. The details can be summarized in two aspects: \
> $(1)$ the uniform stability in [6] is defined as the upper bound of the change of the loss when we REMOVE one sample from the dataset; while in our work, the uniform stability is defined as the upper bound of the change of the loss when we REPLACE one sample of the dataset. \
> $(2)$ As shown in the discussion under the Eq.(7) of [6], if an algorithm has uniform stability $\gamma$ (w.r.t. the exclusion of one sample), then the algorithm has uniform stability $2\gamma$ (w.r.t. the change of one sample). Therefore, we replace the $2\gamma$ factor (w.r.t. the exclusion of one point) in Theorem 12 of [6] with $\gamma$ (w.r.t. the change of one point), leading to the bound in our Theorem 1..
>
> **Q6. Theorem 1 corresponds to [8, Theorem 2], not [8, Theorem 1]**.\
> A: Thanks. We have corrected this typo in the revised version.
>
> **Q7. In Remark 6 “m=1 in few-shot learning”. If m=1 is impossible to split the dataset in support and query. At least should be m=2 but in practice is larger than that**.\
> A: Thanks. We have given a more rigorous statement in our Remark 6 in the revised version.
>
> **Reference**
>
> [4] J. Baxter. A model of inductive bias learning. Journal of Artificial Intelligence Research, 12:149–198, 2000.
>
> [6] O. Bousquet and A. Elisseeff. Stability and generalization. Journal of Machine Learning Research (JMLR), 2:499–526, 2002.
>
> [22] J. Guan and Z. Lu. Task relatedness-based generalization bounds for meta learning. In International Conference on Learning Representations (ICLR), 2022.
>
> [33] A. Maurer. Algorithmic stability and meta-learning. Journal of Machine Learning Research (JMLR), 6:967–994, 2005.
>
> [37] A. Pentina and C. H. Lampert. A PAC-Bayesian bound for lifelong learning. In International Conference of Machine Learning (ICML), pages 991–999, 2014.
>
> [38] J. Rothfuss, V. Fortuin, M. Josifoski, and A. Krause. PACOH: Bayes-optimal meta-learning with PAC-Guarantees. In International Conference on Machine Learning (ICML), pages 9116–9126, 2021.

---

> > ### Comment · Reviewer_E4JU · 2022-08-08
> > **Thank you for the detailed response.**
> >
> > I am partly satisfied with the author response which incorporated some of my suggestions in the updated manuscript. However, I am keeping the weak accept score since the authors did not mention that the fast generalisation bound for PL functions is "deformed" neither in the abstract nor in the contribution section. I would be happy to raise my score if this is done.

---

> > > ### Author Response · Authors · 2022-08-09
> > > **Further Response to the Review by Reviewer E4JU**
> > >
> > > **Q1. The authors did not mention that the fast generalization bound for PL functions is "deformed" neither in the abstract nor in the contribution section. I would be happy to raise my score if this is done**.\
> > > A: Dear Reviewer E4JU,\
> > > Thanks for your multi-round discussions and constructive suggestions. We have mentioned that the fast generalization bound for PL functions is "deformed" in the abstract, contribution, related work, and conclusion section in the updated version (rendered in purple in .pdf).

---

> > > > ### Comment · Reviewer_E4JU · 2022-08-09
> > > > **Thank you for incorporating my feedback in the updated version of the paper.**
> > > >
> > > > I updated the review and increased the score as promised.

---

> > > > > ### Author Response · Authors · 2022-08-09
> > > > > **Thank you for your positive feedback!**
> > > > >
> > > > > We appreciate your support very much!

---

### Official Review · Reviewer_DKCG · 2022-07-06

**Rating:** 6
**Confidence:** 4
**Soundness:** 3 good
**Presentation:** 3 good
**Contribution:** 2 fair

**Summary:**


This paper studies stability and generalization in meta-learning. Specifically, it develops matching upper and lower bounds for non-smooth convex loss function with Holder continuous subgradients, as well as smooth convex and smooth nonconvex functions.

The paper also provides generalization bounds for both independent and dependent episode environments. It shows that in the independent episode environment, the generalization bound of $O(1/\sqrt{n})$ is nearly optimal. And it can achieve $O(1/n)$ with an additional curvature (Polyak-Lojasiewicz) condition of the loss function.


**Questions:**

**Major comments**

1. Questions regarding Table A.2

* Why in Table A.2, there is no comparison with [8,16], which is the most relevant paper that also provides stability-based generalization bounds for meta learning?

* In Table A.2, what is the difference between \gamma_n and \beta_n should be stated more clearly. For example, the uniform stability definitions in those papers are the same or not? If they are the same, why don’t you use the same notation?

* In Table A.2, the last column should be generalization gap bounds instead of transferring error bounds as it does not explicitly include the dependence of the training error \hat{er}(\cdot).


2. Questions regarding the new uniform stability notion

* Definition 2 is a stronger notion of uniform stability compared to single level problems and the definition in [8,16] for meta learning, since it requires changing both K points in the training set and q points in the test set. Why is it necessary to define the uniform stability of meta learning in this way instead of just changing one point in the training or test set?

* Though the new transfer error bound is sharper, it requires the stronger uniform stability definition in Definition 2. Therefore it is not a very fair comparison with existing works such as [8].

3. Questions regarding the notation A(S)(S)

* The notation A(S)(S) is a bit confusing to me, what is the output of hypothesis A(S)(S) should be stated clearly. Is it the per-task (e.g. for the i-th task) hypothesis or the meta model hypothesis (w)?

* Does this S belong to S or it can be any set S? Since I see A(S)(S), A(S)(S^{tr}), and A(S)(S^{tr}_i).

4. Questions regarding remark 1

* In Remark 1, line 236-237, how does Theorem 2 show the importance of good embedding to generalization? To show this, I would expect a term in Eq. (4) that is directly related to the representation or embedding error. I think the authors need to elaborate more on this point.


**Minor comments**

1. Should line 130: $|\partial f(u,z) - \partial f(v,z)| be \|\partial f(u,z) - \partial f(v,z)\|$ instead?

2. What is S^{tr} in Eq. (2) and line 163, is it $S^{tr} = \{ S^{tr}_i\}_{i=1}^n$ or is it just a general notation of S^{tr}_i?

3. In line 163-164, why is S ~ D^m and S ~ D_{\tau} at the same time?

4. In this paper, if I understand correctly, the number of episodes is equal to the number of tasks in meta training. This should be stated more clearly at the beginning of the paper, for example, move Algorithm 1 to the main text as it is important to understand the settings of the analysis.

5. Should line 161-162: $\mathbf{D}_{\tau}$ be  $\mathbf{D}_{\tau}^m$?

6. It would be more interesting to add additional experiments on real-world data such as for few-shot image classification tasks.

7. It would be better if the authors could add a theoretical curve of the generalization gap v.s. the number of tasks in Figure 1 and Figure 2.

8. In Remark 1, line 232-234, it is stated that [29, 30] have stability bounds that depend on the empirical risk, which I do not agree since they obtain dependence on either the population risk or the expected empirical risk, which is not directly the empirical risk minimized during optimization without expectation over the data sample and algorithm. Therefore, their theoretical results do not imply small training error leads to a small generalization error.

9. Grammar

* Line 160: as follow -> as follows

* Line 423: stability notations -> stability notions



**Strengths And Weaknesses:**

**Strengths**

1. This is the first work that captures the effect of dependent episodes in generalization ability, which explains the empirical discovery that meta learning trained with independent episodes generalizes better than with dependent episodes.

2. The consideration of the weakest curvature condition, Polyak-Lojasiewicz condition, instead of the strong-convexity condition for O(1/n) rate of generalization is interesting.

3. This work develops stability of meta learning for both smooth and non-smooth loss functions. And conduct experiments with examples in both conditions to verify their theorems.

4. The paper provides matching upper and lower bounds for stability to show the upper bound is tight.

**Weaknesses**

1. The comparison to [8,16] is not clear, which is the most relevant papers that also provide stability-based generalization bounds for meta learning.

2. The techniques to obtain stability in non-smooth convex loss are not new as it is already established in single-level problems [30]. And the technique for lower bound is also not new. The paper directly assumes continuity or smoothness or convex assumptions for the outer level function w.r.t. the meta parameter w, therefore, the analysis of single-level ERM can be directly applied without many challenges that are uniquely caused by the bi-level (compositional) structure of the problem.


3. The authors sometimes exchange the notions of “transfer error” and “generalization gap” in the paper. For example, in lines 302, 322, and 344, Table A.2, should the “transfer error” be the “generalization gap”?

4. The stability bound in Theorem 4 requires step sizes $\eta_j = O(\frac{1}{j})$, while the convergence of a smooth nonconvex loss function for compositional or bilevel problems requires outer level step size is at least $\eta_j = O(\frac{1}{\sqrt{j}})$ (see e.g. [1,2,3]). What is the step size used for the experiments presented in Section 6, where both the training error and generalization gap converge? This phenomenon should receive more explicit discussions.


5. Since the experiment settings belong to a non-convex non-smooth loss function, it is not covered by the theoretical results that include smooth convex, smooth non-convex, and non-smooth convex functions. It would be better to conduct some simulations with the loss functions under the Assumptions used in the theoretical results to verify the theoretical rates more explicitly. Therefore, the experiments in Section 6 do not serve the purpose of verifying the theoretical claims. The observation that a non-smooth non-convex loss function has similar convergence rates in the generalization gap as the smooth non-convex loss function suggests further study and should be pointed out clearly in the paper.

6. Some discussions are not clear to me, as detailed in “Questions”.

[1] Closing the gap: tighter analysis of alternating stochastic gradient methods for bilevel problems

[2] On the Convergence Theory of Gradient-Based Model-Agnostic Meta-Learning Algorithms

[3] Theoretical Convergence of Multi-Step Model-Agnostic Meta-Learning

---

> ### Author Response · Authors · 2022-08-01
> **Response to the Review by Reviewer DKCG (1)**
>
> **Q1. The comparison to [8,16] is not clear.** \
> A: Thanks. Our explanations are two-fold: \
> $(1)$ For the comparison to [8]: the detailed comparison to [8] is listed in our Remark 4 in the main paper. Besides, in Table A.2 of Appendix A, we also compare our bound with the bound from [8] (the reference [6] in the appendix is actually the reference [8] in the main paper). \
> $(2)$ For the comparison to [16]: it is still hard for us to directly compare our bounds with that of [16], the reasons are as follows: $(i)$ We focus on different bounding objectives. Our work aims to bound the transfer error over the novel task (under the task environment assumption), whereas [16] aims to bound the (expected) excess risk over the novel task (without the task environment assumption, see its Corollary 2). $(ii)$ The generalization bounds hold with different forms. The bounds in our Theorems 5-7 all hold with high probability, but the generalization bounds in [16] (i.e. the bound on the gap between the expected multi-task error and empirical multi-task error in its Theorem 1, as well as the bound on the excess risk on the novel task in its Corollary 2) hold in expectation (w.r.t. all training samples). $(iii)$ We take different assumptions of the loss function. In Assumption 1 of [16], the authors assume the loss function satisfy 4 conditions: strong convexity, Lipschitzness, smoothness and Hessian Lipschitzness. But our work only takes one or two conditions to derive stability for meta learning algorithm. Consider the aforementioned reasons, we think it is not suitable to directly compare our in-probability generalization bound with the in-expectation bound of [16]. We have added such explanations in Remark A.1 in Appendix A in the revised version (the reference [16] in the main paper is the reference [7] in the appendix).
>
> **Q2. The techniques to obtain stability in non-smooth convex loss are not new as it is already established in single-level problems [30].** \
> A: Thanks. The techniques to obtain stability in non-smooth convex loss are originated from [1] (for convex and Lipschitz loss), and in this work we extend such techniques to the convex Holder smooth setting. Meanwhile, we also use the techniques from [30] to derive uniform argument stability for non-smooth convex loss (see our Theorem D.1 and Theorem D.2 in Appendix D.2.2), and compare the tightness of the stabilities obtained with different techniques in our Remark 3.
>
> **Q3. In lines 302, 322, and 344, Table A.2, should the “transfer error” be the “generalization gap**”? \
> A: Sorry for the confusion. Although transfer error bound and the bound on the generalization gap are equivalent to some extent, “transfer error bound” always includes the empirical error term, whereas “the bound on the generalization gap” does not involve the empirical error term. We have clarified these notations in the revised version.
>
> **Q4. What is the step size used for the experiments presented in Section 6?** \
> A: Thanks. We follow the same experimental setting in [8,20], and the initial learning rate in these works is set as $0.001$.
>
> **Q5. Since the experiment settings belong to a non-convex non-smooth loss function, it is not covered by the theoretical results**. \
> A: Thanks. Although it is hard to validate the smoothness of the neural network model in our experiment, we assume $l_{2}$ loss is smooth and $l_{1}$ loss is non-smooth (w.r.t. parameter $w$). Therefore, our experiments can verify the generalization behavior of non-convex smooth loss. For non-convex non-smooth loss function $l_{1}$, deriving its stability is still difficult and serves one of our ongoing research. We conduct experiments with $l_{1}$ loss to see whether there exists difference of convergence performance between smooth and non-smooth functions.
>
> **Q6. In Table A.2, what is the difference between $\gamma_n$ and $\beta_n$?** \
> A: Sorry for the confusion. Our explanations are two-fold: \
> $(1)$ the $\gamma_{n}$ is Table A.2 represents the uniform stability of meta algorithms, where the subscript $n$ means the number of training tasks; the $\gamma_{m}$ represents the uniform stability of inner-task algorithm,  where the subscript $m$ means the number of samples per task. \
> $(2)$ $\beta_n$ represents the uniform argument stability defined in our Definition 3,  and the subscript $n$ means the number of training tasks. We have added more explanations in the caption of Table A.2 in the revised version.

---

> ### Author Response · Authors · 2022-08-01
> **Response to the Review by Reviewer DKCG (2)**
>
> **Q7. Why not define the uniform stability of meta learning by just changing one point in the training or test set?** \
> A: Thanks. The reasons for defining the uniform stability of meta learning algorithms in this way lie in two aspects: \
> $(1)$ Under the task environment assumption, we can regard the whole dataset in one task as a training sample, then for a meta algorithm whose input is the datasets from all training tasks, the algorithmic stability of a meta algorithm should be defined by changing the dataset corresponding to one task. \
> $(2)$ Such uniform stability definition method is originated from [33], and is developed by [8]. We follow the work of [8,33] to develop improved bounds for fair comparisons, and also point out the limitations of such stability notions for meta learning in our Remark 5.
>
> **Q8. Though the new transfer error bound is sharper, it requires the stronger uniform stability definition than [8]**. \
> A: Thanks. Our explanations are two-fold: \
> $(1)$ Actually, our improved stability-based transfer error bounds in Theorem 5 and 6 are not only applicable to uniformly argument stable meta learning algorithms, but also applicable to uniformly stable meta learning algorithms. \
> $(2)$ The reason for the improvements of our results over [8] is not the use of a stronger definition, but the key observation that single-task learning and episodic S/Q meta-learning are essentially equivalent. Therefore, we can apply recent fast-rate stability-based bounds in single-task learning to meta learning to give improved meta learning bounds. Therefore, the comparisons with [8] are fair.
>
> **Q9. What is the output of hypothesis** A(S)(S)?\
> A: $\mathbf{A}(\mathbf{S})$ is an inner-task algorithm. Given the training dataset $S$ associated with one task, the algorithm $\mathbf{A}(\mathbf{S})$ will output a hypothesis $\mathbf{A}(\mathbf{S})(S)$ suitable for that task. Such hypothesis always contains two parts: (1) An embedding that is shared across different tasks; (2) A prediction function (on the top of features extracted by the embedding) suitable for that task.
>
> **Q10. In Remark 1, how does Theorem 2 show the importance of good embedding to generalization?** \
> A: Thanks. A good embedding means that the model has a good initialization and can achieve low empirical errors at the first several optimization steps, and hence can have a small stability. We have added more explanations in Remark 1.
>
> **Q11. Is S^{tr} in Eq. (2) just a general notation of S^{tr}_i?** \
> A: $S^{tr}$ in Eq. (2) is the support set in one training task. It is just a general notation of $S^{tr}_i$.
>
> **Q12. In line 163-164, why is S ~ D^m and S ~ D_{\tau} at the same time?** \
> A: Our explanations are two-fold: \
> $(1)$ First, each sample in $ S $ is drawn independently according to the probability measure $D$, therefore $S$ can be regarded as sampled according to the measure $D^{m}$; \
> $(2)$ In meta-learning, under the task environment assumption, the probability measure $D$ is regarded as a random variable and is sampled from the environment $\tau$, therefore we have $D \sim \tau$. Combining (1) and (2), in meta learning setting we have $S \sim D^{m}, D\sim \tau$, and we can write $S \sim \mathbf{D}_{\tau}$ for simplicity (see the formal definition in lines 161-162).
>
> **Q13. Should line 161-162: $\mathbf{D}_{\tau}$  be $\mathbf{D}_{\tau}^m$?** \
> A: No. $\mathbf{D}_{\tau}$ in line 161-162 is actually a probability measure over the space $\mathcal{Z}^{m}$.
>
> **Q14. It would be more interesting to add additional experiments on real-world data such as for few-shot image classification tasks.** \
> A: Thanks. Actually we have conducted experiments on miniImageNet for few-shot image classification tasks as in [8]. However, we found that the convergence performance of generalization gap was not so satisfactory. We thought maybe it is because the classification episodes cannot be regarded as “sampled independently from one environment”. Therefore, we only conduct simulation experiments on regression tasks where the parameter corresponding to each task can be guaranteed to be sampled independently from a task distribution.
>
> **Q15. It would be better to add a theoretical curve of the generalization gap v.s. the number of tasks in Figures 1-2.** \
> A: Our explanations are two-fold: \
> $(1)$ Since it is hard to approximately estimate the Lipschitzness, smoothness and boundedness constants of the loss function, it is difficult to directly give the order of algorithmic stability $\beta$. \
> $(2)$ We cannot estimate how large our generalization bound can be (i.e., we have no idea what the exact multiplicative factor is in our optimal bound $O(1/\sqrt{n})$). Therefore, we are unable to add a theoretical curve of the generalization gap in Figures 1-2.
>
> **Q16. In Remark 1, the stability bounds in [29, 30] depend on either the population risk or the expected empirical risk.** \
> A: Thanks. We have refined our Remark 1 in the revised version.

---

> ### Comment · Reviewer_DKCG · 2022-08-07
> **Thanks for the rebuttal!**
>
> Dear authors,
>
> Thanks for the rebuttal. Most of my questions have been answered except Q4.
> My question was that if the stepsizes required in the analysis contradict with those in convergence analysis of meta-learning algorithms. If so, do the stepsizes in the experiments follow $1/j$ or $1/\sqrt{j}$? It will not affect my rating, but clarifying would be great.

---

> > ### Author Response · Authors · 2022-08-07
> > **Further Response to the Question by Reviewer DKCG**
> >
> > **Q1. My question was that if the stepsizes required in the analysis contradict with those in convergence analysis of meta-learning algorithms. If so, do the stepsizes in the experiments follow $O(1/j)$ or $O(1/\sqrt{j})$?** \
> > A: Dear Reviewer DKCG, \
> > Thanks for your responses. Our explanations are two-fold: \
> > $(1)$ For the convergence guarantee of MAML in our Theorem 4: It seems like that both reference [2] and [3] require the step size $\eta_{j}  \in (0, \frac{1}{G})$ for $G$-smooth functions to guarantee the convergence of gradient based MAML algorithms (see Theorem 5.12 in [2] and Theorem 5 in [3]). Reference [1] also requires the step size $\eta_{j}  \leq \min \lbrace O(\frac{1}{\sqrt{j}}), \frac{1}{G}\rbrace$ to guarantee the convergence of MAML (see Eq.(12) in Theorem 1 in [1]). Therefore, it seems like that our step size $\eta_{j}=O(\frac{1}{jG})$ in our Theorem 4 satisfies the requirements in [1,2,3] (i.e. $\eta_{j}=O(\frac{1}{jG}) \leq \frac{1}{G}$ and $\eta_{j}=O(\frac{1}{jG}) \leq\frac{1}{\sqrt{j}}$) , and our step size can guarantee the convergence performance. \
> > $(2)$ For the experimental setting: on one hand, for the fair comparisons with existing works, we follow the step size setting as in reference [4,5] where the initial learning rate of meta learning algorithms is set as $0.001$, and we use the function torch.optim.lr_scheduler.StepLR(step_size=20, gamma=0.5) to decrease our learning rate every 20 training epochs. On the other hand, sine it is hard to approximately estimate the order of the smoothness constant $G$ of the $G$-smooth loss function, it is infeasible to set step sizes $\eta_{j}=O(\frac{1}{jG})$ in practice. Therefore, the step size in our experimental setting does not rigorously follow the one in our theoretical setting. However, our step size in practice may also cause convergence result of MAML, since our step sizes in experiment may be small enough (i.e.  less than $\frac{1}{G}$) to guarantee the convergence.
> >
> > **Reference**
> >
> > [1] Closing the gap: tighter analysis of alternating stochastic gradient methods for bilevel problems
> >
> > [2] On the Convergence Theory of Gradient-Based Model-Agnostic Meta-Learning Algorithms
> >
> > [3] Theoretical Convergence of Multi-Step Model-Agnostic Meta-Learning
> >
> > [4] A closer look at the training strategy for modern meta-learning.
> >
> > [5] Model-agnostic meta-learning for fast adaptation of deep networks.

---

### Official Review · Reviewer_Skjh · 2022-07-11

**Rating:** 7
**Confidence:** 4
**Soundness:** 4 excellent
**Presentation:** 3 good
**Contribution:** 3 good

**Summary:**

This manuscript provide fine-grained analysis of stability and generalization for modern meta learning algorithms by considering more general situations including -Holder continue convex, smooth convex and smooth non-convex functions. First, the authors give the lower and upper bounds of the uniform argument stability for -Holder continue convex functions and smooth convex functions respectively to show that meta learning algorithms in the smooth convex case is more stable than that in the non-smooth convex case. Second, a tighter stability bound of  than the existing bound  is proved. Then, to show the advantage of S/Q episodic strategy for meta learning over traditional ERM strategy, the authors develop a near-optimal high-probability generalization bound , and further improve the bound to  by considering Polyak-Łojasiewicz condition. Finally, a generalization bound for meta learning with dependent episodes is given.

**Questions:**

1. For the upper bound of the uniform argument stability in Theorem 3 in line 259 of the manuscript, there is not the prove of it in Appendix. Similarly, the upper bound in line 267 of the manuscript is also not proved.
2. In line 293 of the manuscript, the authors show the result of Theorem 1 in reference [8] is  and , that is, the bound is , which is the same as the bound of Theorem 5 in this manuscript. Why does the authors think the result of this manuscript is better that that in reference [8]?


**Limitations:**

It seems no potential negative societal impact, sine this paper just considers theoretical properties of meta learning.

**Strengths And Weaknesses:**

Pros
1.The symbol description of the manuscript is relatively detailed.
2.The manuscript comprehensively considers various function settings, and gives relatively tight results.
3.The results of this manuscript are compared with the results of many previous works, and the authors show the situations where these results are better in detail.

Cons
1.The coefficient of Theorem 1 cited in this manuscript is slightly different from that in the reference [6].
2.In Appendix D1.1, Lemma D.1 does not mention the precondition of  which is written in the subtitle. So what is this precondition for?
3.At the beginning of the proof of Theorem 2 (line 71 in Appendix) and Theorem 3 (line 223 in Appendix) in this manuscript, the nonexpansiveness of projection operator is used. Whether the nonexpansiveness needs to be proved?

In summary, I believe this paper is interest. It provides some tighter stability bounds and generalization bounds for meta learning.

---

> ### Author Response · Authors · 2022-08-01
> **Response to the Review by Reviewer Skjh**
>
> **Q1. The coefficient of Theorem 1 cited in this manuscript is slightly different from that in the reference [6]**? \
> A: Thanks for pointing this out. The reason for the difference is that the uniform stability notion in our Theorem 1 is slightly different from the uniform stability notion in reference [6]. The details are as follows: \
> $(1)$ the uniform stability in [6] is defined as the upper bound of the change of the loss when we REMOVE one sample from the dataset; while in our work, the uniform stability is defined as the upper bound of the change of the loss when we REPLACE one sample of the dataset. \
> $(2)$ As shown in the discussion under the Eq.(7) of [6], if an algorithm has uniform stability $\gamma$ (w.r.t. the exclusion of one sample), then the algorithm has uniform stability $2\gamma$ (w.r.t. the change of one sample). Therefore, we replace the $2\gamma$ factor (w.r.t. the exclusion of one point) in Theorem 12 of [6] with $\gamma$ (w.r.t. the change of one point), leading to the bound in our Theorem 1.
>
> **Q2. In Appendix D1.1, Lemma D.1 does not mention the precondition of which is written in the subtitle. So what is this precondition for?**  \
> A: Sorry for the confusion. Our Lemma D.1 truly holds without the precondition $T > n$ written in the subtitle. The precondition $T > n$ in the subtitle indicates that the stability bound in our Lemma D.1 is more suitable for the case when $T > n$. When $T \leq n$, we can derive a sharper stability bound in Lemma D.2 for convex Holder smooth function. We have added more explanations above Lemma D.1 in the revised version.
>
> **Q3. Whether the nonexpansiveness (line 71 in Appendix and line 223 in Appendix) of the projection operator needs to be proved**? \
> A: Thanks. The proof of the nonexpansiveness of projection operator in Euclidean space is a little lengthy and unrelated to the theoretical results in this work. Therefore we omit the detailed proof of the nonexpansiveness and refer interested readers to Proposition 4.4 in the book “Convex Analysis and Monotone Operator Theory in Hilbert Spaces”.
>
> **Q4. The upper bound of the uniform argument stability in Theorem 3 in line 259 of the manuscript, and the upper bound in line 267 of the manuscript are not proved.** \
> A: Thanks. The proof for the upper bound of the uniform argument stability in Theorem 3 (in line 259 of the manuscript) is deferred to line 229-233 in Appendix D.3. The proof for the upper bound in line 267 of the manuscript is deferred to line 237-240 in Appendix D.4.
>
> **Q5. In line 293 of the manuscript, the authors show the result of Theorem 1 in reference [8] is the same as the bound of Theorem 5 in this manuscript. Why do the authors think the result of this manuscript is better than that in reference** [8]? \
> A: Thanks. Consider a scenario where the algorithmic stability $\gamma$ of SGD has the order $O(1/\sqrt{n})$ (such example can be found in the discussion under Eq.(2) in [19]), then the bound $O(\gamma\sqrt{n} + M/\sqrt{n})$ in Theorem 1 of reference [8] will become vacuous (i.e., $O(\gamma\sqrt{n} + M/\sqrt{n})=O(1)$), while our bound of $O(\gamma\ln{n} + M/\sqrt{n}) )$ in Theorem 5 has the order of $O(\ln{n}/\sqrt{n})$ and still has an asymptotic guarantee. Therefore, our result in Theorem 5 is better than that in reference [8]. More explanations for our improvements can also be found in our Remark 4.
>
> **Reference**
>
> [6] O. Bousquet and A. Elisseeff. Stability and generalization. Journal of Machine Learning Research (JMLR), 2:499–526, 2002.
>
> [8] J. Chen, X. Wu, Y. Li, Q. LI, L. Zhan, and F. Chung. A closer look at the training strategy for modern meta-learning. In Conference on Neural Information Processing Systems (NeurIPS), pages 396–406, 2020.
>
> [19] V. Feldman and J. Vondrák. High probability generalization bounds for uniformly stable algorithms with nearly optimal rate. In Conference on Learning Theory (COLT), pages 1270–1279, 2019.

---

### Official Review · Reviewer_rSh4 · 2022-07-25

**Rating:** 6
**Confidence:** 4
**Soundness:** 3 good
**Presentation:** 3 good
**Contribution:** 2 fair

**Summary:**

This paper studies the generalization of meta-learning algorithms. In particular, this paper has three contributions:
1- Extending the stability analysis for convex smooth functions to convex Holder-smooth functions (similar to [30] for single task)
2- Improving the stability analysis of nonconvex functions
3- High probability generalization bounds

In particular, the authors show that the meta learning algorithms are less stable in the nonsmooth convex case, compared to smooth convex case.

**Questions:**

I would appreciate it if the authors discuss how they could potentially study the effect of $m$ in generalization of meta-learning  algorithms for smooth and nonsmooth functions.

**Limitations:**

Yes.

**Strengths And Weaknesses:**

Overall, I find the paper and its results interesting to the community. My main concern is that the algorithmic stability is defined in a way that the whole dataset corresponding to a task changes. This makes the analysis simpler (as it would be closer to the single task case), but the results will not be tight with respect to the number of samples per task (m), as the authors point out in Remark 5.

---

> ### Author Response · Authors · 2022-08-01
> **Response to the Review by Reviewer rSh4**
>
> **Q1. My main concern is that the algorithmic stability is defined in a way that the whole dataset corresponding to a task changes. The results will not be tight with respect to the number of samples per task (m), as the authors point out in Remark 5**.\
> A: Thanks. There are two main reasons for our generalization bounds unrelated to the number of samples per task ($m$):\
> $(1)$ We rely on a basic assumption that the tasks are sampled from the same environment. \
> $(2)$ Our algorithmic stability is defined in a way that the whole dataset corresponding to a task changes. \
> Actually, only under the task environment assumption, can we view the dataset in each training task as a training sample, treat single-task learning and episodic meta learning equally, and derive an optimal bound of $O(1/\sqrt{n})$ for meta learning via algorithmic stability analysis. Therefore, our bound also reveals the limitation of the task environment assumption.
>
> **Q2. I would appreciate it if the authors discuss how they could potentially study the effect of m in generalization of meta-learning algorithms for smooth and nonsmooth functions**. \
> A: Thanks for your comments. Our explanations are two-fold: \
> $(1)$ Under the task environment assumption: actually we can extend the algorithmic stability notions in single-task learning (e.g. uniform stability [6], uniform argument stability [2,32], on-average stability [29], on-average model stability [30]) to the episodic meta learning setting by defining an algorithmic stability in a way that the whole dataset corresponding to a task changes (see our Definition 2 and Definition 3). However, no matter which stability notion we use, our Remark 5 tells us that the stability-based transfer error bound will not be tighter than $O(1/\sqrt{n})$. Therefore, to derive sharper transfer error bound (e.g. of $O(1/\sqrt{nm})$) for meta learning under the task environment assumption, we should leverage the tools of other theories (e.g. model-capacity theory in [5]), instead of the tool of algorithmic stability analysis. \
> $(2)$ Without the task environment assumption: without such assumption, we cannot define the transfer error of a meta  learning algorithm on the novel task, so we should focus on the excess risk bound on the novel task. In this case, we may define a more elaborate algorithmic stability notion in a way that the part (not the whole) of dataset in a task change, and may derive a sharper bound that is related to the number $m$ of samples per task (like the expected multi-task error bound of $O(\frac{1}{nm})$ in [16]).
>
> **Reference**
>
> [2] R. Bassily, V. Feldman, C. Guzmán, and K. Talwar. Stability of stochastic gradient descent on nonsmooth convex losses. In Conference on Neural Information Processing Systems (NeurIPS), 2020.
>
> [5] S. Ben-David and R. Schuller. Exploiting task relatedness for mulitple task learning. In Conference on Learning Theory (COLT), pages 567–580, 2003.
>
> [6] O. Bousquet and A. Elisseeff. Stability and generalization. Journal of Machine Learning Research (JMLR), 2:499–526, 2002.
>
> [16] A. Fallah, A. Mokhtari, and A. E. Ozdaglar. Generalization of model-agnostic meta-learning algorithms: Recurring and unseen tasks. In Conference on Neural Information Processing Systems (NeurIPS), 2021.
>
> [29] I. Kuzborskij and C. H. Lampert. Data-dependent stability of stochastic gradient descent. In International Conference on Machine Learning (ICML), pages 2820–2829, 2018.
>
> [30] Y. Lei and Y. Ying. Fine-grained analysis of stability and generalization for stochastic gradient descent. In International Conference on Machine Learning (ICML), pages 5809–5819, 2020.
>
> [32] T. Liu, G. Lugosi, G. Neu, and D. Tao. Algorithmic stability and hypothesis complexity. In International Conference on Machine Learning (ICML), pages 2159–2167, 2017.

---

> > ### Comment · Reviewer_rSh4 · 2022-08-08
> > **Thank you for the response**
> >
> > I would like to thank authors for their response. After reading other reviews and responses, I have decided to keep my score as is.

---

> > > ### Author Response · Authors · 2022-08-09
> > > **Thank you for your positive feedback!**
> > >
> > > We appreciate your support very much!

---

### Author Response · Authors · 2022-08-01
**Summary for Revision**

We thank all reviewers for their detailed reading and constructive comments. In the revision, we made the following major changes (rendered in purple in .pdf) regarding the reviewers' concerns:

1. A new remark (Remark A.2 in Appendix A) explaining how we could potentially derive a sharper generalization bound w.r.t. $m$ for meta-learning algorithms (In response to the question by **Reviewer rSh4**);

2. Clarification on the precondition of Lemma D.1 in Appendix D.1.1 (As suggested by **Reviewer Skjh**);

3. A new remark (Remark A.1 in Appendix A) explaining the differences between our high-probability generalization bounds for meta learning and the in-expectation generalization bounds in [Fallah et al. NeurIPS2021]. (In response to Weakness 1 by **Reviewer DKCG**);

4. Fixing important typos (e.g. use “Bounds on Generalization Gap” instead of “Transfer Error Bounds”) and giving more explanations for different algorithmic stability notions in Table A.2 (In response to Question 1 by **Reviewer DKCG**);

5. Giving a more rigorous statement in our Remark 1 (In response to Minor Comments 8 by **Reviewer DKCG**);

6. Clarifying the “deformed” transfer error bound in our Theorem 6 (In response to Question 1 by **Reviewer E4JU**);

7. Clarifying the contribution (in our Introduction and Remark 6) of giving more comprehensive comparisons between the bounds for S/Q meta-learning and the bounds for ERM meta-learning (In response to Question 2 by **Reviewer E4JU**);

8. Clarifying the citation of our Lemma E.1, including log terms inside $O(\cdot)$ throughout the paper, and fixing important typos in Remark 6 (In response to Questions 3-4 and Minor Question 4 by **Reviewer E4JU**).

Besides the above major changes, we also fixed other minor typos.

---

### Meta-Review · Area_Chair_pJCf · 2022-08-20

**Recommendation:** Accept
**Confidence:** Certain

**Metareview:**

The reviewers and AC are in agreement that this paper is a solid work, and its contributions are significant. The theoretical results of this paper advance the theory of meta-learning, and, in particular, the provided generalization guarantees are strong. All reviewers were satisfied with the responses provided by the authors and even one of the reviewers increased their score. Overall, this is a good paper and my recommendation is "Accept".

AC

**Award:**

No

---

### Decision · Program_Chairs · 2022-09-14

Accept